# Reduced Tolerogenic Program Death-Ligand 1-Expressing Conventional Type 1 Dendritic Cells Are Associated with Rapid Decline in Chronic Obstructive Pulmonary Disease

**DOI:** 10.3390/cells13100878

**Published:** 2024-05-20

**Authors:** Kuan-Yuan Chen, Wei-Lun Sun, Sheng-Ming Wu, Po-Hao Feng, Chiou-Feng Lin, Tzu-Tao Chen, Yueh-Hsun Lu, Shu-Chuan Ho, Yueh-Hsi Chen, Kang-Yun Lee

**Affiliations:** 1Graduate Institute of Clinical Medicine, College of Medicine, Taipei Medical University, Taipei 110, Taiwan; 14388@s.tmu.edu.tw (K.-Y.C.); 09330@s.tmu.edu.tw (T.-T.C.); 2Division of Pulmonary Medicine, Department of Internal Medicine, Shuang Ho Hospital, Taipei Medical University, New Taipei City 235, Taiwan; brontesun@gmail.com (W.-L.S.); chitosan@tmu.edu.tw (S.-M.W.); fengpohao@tmu.edu.tw (P.-H.F.); shu-chuan@tmu.edu.tw (S.-C.H.); kazephirine@gmail.com (Y.-H.C.); 3Division of Pulmonary Medicine, Department of Internal Medicine, School of Medicine, College of Medicine, Taipei Medical University, Taipei 110, Taiwan; 4TMU Research Center of Thoracic Medicine, Taipei Medical University, Taipei 110, Taiwan; 5Department of Medical Research, Shuang Ho Hospital, Taipei Medical University, New Taipei City 235, Taiwan; 6Department of Microbiology and Immunology, School of Medicine, College of Medicine, Taipei Medical University, Taipei 110, Taiwan; cflin2014@tmu.edu.tw; 7Department of Radiology, Shuang-Ho Hospital, Taipei Medical University, New Taipei City 235, Taiwan; d755111001@tmu.edu.tw; 8School of Respiratory Therapy, College of Medicine, Taipei Medical University, Taipei 110, Taiwan

**Keywords:** programmed death-ligand 1 (PD-L1), conventional type 1 dendritic cells (cDC1s), chronic obstructive pulmonary disease (COPD), rapid decline phenotype

## Abstract

Background: Chronic obstructive pulmonary disease (COPD) is characterized, at least in part, by autoimmunity through amplified T helper 1 and 17 (Th1 and Th17) immune responses. The loss of immune tolerance controlled by programmed death-ligand 1 (PD-L1) may contribute to this. Objectives: We studied the tolerogenic role of PD-L1^+^ dendritic cells (DCs) and their subtypes in relation to specific T cell immunity and the clinical phenotypes of COPD. Methods: We used flow cytometry to analyze PD-L1 expression by the DCs and their subtypes in the peripheral blood mononuclear cells (PBMCs) from normal participants and those with COPD. T cell proliferation and the signature cytokines of T cell subtypes stimulated with elastin as autoantigens were measured using flow cytometry and enzyme-linked immunosorbent assays (ELISA), respectively. Measurement and main results: A total of 83 participants were enrolled (normal, *n* = 29; COPD, *n* = 54). A reduced PD-L1^+^ conventional dendritic cell 1 (cDC1) ratio in the PBMCs of the patients with COPD was shown (13.7 ± 13.7%, *p* = 0.03). The decrease in the PD-L1^+^ cDC1 ratio was associated with a rapid decline in COPD (*p* = 0.02) and correlated with the CD4^+^ T cells (*r* = −0.33, *p* = 0.02). This is supported by the NCBI GEO database accession number GSE56766, the researchers of which found that the gene expressions of *PD-L1* and *CD4*, but not *CD8* were negatively correlated from PBMC in COPD patients (r = −0.43, *p* = 0.002). Functionally, the PD-L1 blockade enhanced CD4^+^ T cell proliferation stimulated by CD3/elastin (31.2 ± 22.3%, *p* = 0.04) and interleukin (IL)-17A production stimulated by both CD3 (156.3 ± 54.7, *p* = 0.03) and CD3/elastin (148 ± 64.9, *p* = 0.03) from the normal PBMCs. The PD-L1 blockade failed to increase IL-17A production in the cDC1-depleted PBMCs. By contrast, there was no significant change in interferon (IFN)-γ, IL-4, or IL-10 after the PD-L1 blockade. Again, these findings were supported by the NCBI GEO database accession number GSE56766, the researchers of which found that only the expression of *RORC*, a master transcription factor driving the Th17 cells, was significantly negatively correlated to PD-L1 (r = −0.33, *p* = 0.02). Conclusions: Circulating PD-L1^+^ cDC1 was reduced in the patients with COPD, and the tolerogenic role was suppressed with susceptibility to self-antigens and linked to rapid decline caused by Th17-skewed chronic inflammation.

## 1. Introduction

Chronic obstructive pulmonary disease (COPD) is the third leading cause of mortality globally [1]. Smoking is the most common contributor to COPD, which can lead to persistent airway inflammation even after smoking cessation [2,3,4]. The underlying mechanism of self-perpetuating airway-to-lung parenchyma inflammation is still unclear but the macrophage, neutrophil, and B and T lymphocyte inflammatory responses intensify as COPD progresses [3,5]. Among those with T cell immunity, there appear to be CD8^+^ cytotoxic T (Tc), CD4^+^ T helper 1 (Th1), and CD4^+^ T helper 17 (Th17) cells predominant in COPD, which might sustain macrophagic and neutrophilic inflammation [6,7]. Activated CD8^+^ Tc cells produce cytotoxic granule enzymes, and the Th1 cells activate the macrophages to release matrix metalloelastases (MMPs), which lead to apoptosis of alveolar cells and the degradation of the extracellular matrix of lung tissues, aggravating COPD [8,9]. The Th17 cells contribute to the pathogenesis of COPD mainly by producing IL-17A, which facilitates neutrophilic inflammation in the small airways and induces lymphoid neogenesis formation by B cells attracting the chemokine C-X-C motif ligand (CXCL) 12 during the progression of COPD [10,11]. To complicate matters, the lung tissue components destructed by the CD8^+^ Tc and Th1 cells might release auto-antigens, such as elastin peptides, and drive lymphoid neogenesis in a similar way to autoimmune responses [12]. In support of this theory, cigarette smoke could induce Th1 and Th17 immune responses to autoantigen elastin peptides and has been associated with bronchitis and emphysema in both human and mice COPD studies [13,14].

The dysregulation of the inhibitory immune checkpoints has been shown to cause autoimmune diseases [15]. The programmed death-1 (PD-1)/programmed death-ligand 1 (PD-L1) axis is a critical inhibitory immune checkpoint but few studies have been conducted on autoimmunity phenomena in COPD, as mentioned above. Furthermore, COPD is a heterogeneous disease, with a specialized clinical phenotype related to treatment and prognosis [16]. Currently, no research has been performed on the role of the PD-1/PD-L1 axis in different COPD phenotypes, particularly in its relationship with the autoimmune feature triggered by Th17 immunity.

Dendritic cells (DCs), a major link between innate and adaptive immunity, are well-equipped to drive the pathogenesis of COPD from the early to end stages [17]. There are three main subtypes of human DCs, including type 1 and type 2 classical/conventional DCs (cDC1 and cDC2, respectively) and plasmacytoid DCs (pDCs), and both have shared and distinct functions [18]. Whilst cDC1 is able to trigger a Th1 immune response and cross-present antigens related to apoptotic cells, cDC2 primarily produces pro-inflammatory chemokines and recruits inflammatory cells [17]. In mice lungs, CD103^+^ DCs, the equivalent of human cDC1, predominantly elicit Th1 and Th17 responses, whereas CD11b^high^ lung dendritic cells (LDCs) primarily provoke a Th2 response [19]. DCs are also tolerogenic. In addition to human cDC2, mouse intestinal mucosal CD103^+^ DCs play a crucial role in tolerance [20]. The relative inefficiency of those DCs to mediate Th17 cell differentiation could be reversed by the retinoic acid receptor antagonist LE135 [21]. It is not clear whether PD-L1 plays a role in tolerogenic DCs, particularly in the different subtypes, and neither is its clinical relevance.

We hypothesized that impairment in PD-L1-mediated immune tolerance impacts the pathogenesis of COPD. During an initial screen test using PBMCs from normal subjects and patients with COPD, we found the PD-L1 expression level in COPD was decreased in the DCs, particularly in cDC1. Then, we sophisticatedly studied the clinical relevance of this reduction and linked it to CD4^+^ T cell differentiation. In vitro experiments were also conducted to test the functional role of PD-L1.

## 2. Methods

### 2.1. Study Population

A total of 83 participants aged 40–80 years were enrolled (COPD, *n* = 54; healthy, *n* = 29) in this study. Some patients with COPD had undergone high-resolution computed tomography (HRCT) at Shuang Ho Hospital, Taipei Medical University (New Taipei, Taiwan), between March 2015 and February 2021. The COPD severity was based on the Global Initiative for Chronic Obstructive Lung Disease (GOLD) guidelines. The patients were in a stable condition without acute exacerbation for the previous 3 months, which is defined as not needing to be admitted, not being administered antibiotics or systemic oral corticosteroid therapy, and there being no change in their respiratory symptoms. The normal subjects were defined as those without any known medical diseases, and they had not experienced any acute infections, including upper airway infections, in the previous 3 months. Full informed consent was obtained from all the participants before sample collection. The study protocol was approved by the Taipei Medical University Joint Institutional Review Board (N201902021). All experiments were performed in accordance with the relevant guidelines and regulations.

### 2.2. Classification of COPD Phenotype

#### 2.2.1. Rapid Decline

We defined a loss of forced expiratory volume in one second (FEV1) ≥ 60 mL/year as the rapid decline in lung function in COPD [22]; therefore, the patients’ lung function had to be tracked for at least one year. Based on the definition, a total of 9 rapid decliners and 35 non-rapid decliners were included in this study. Their lung function was studied for an average of 3.5 ± 1.4 years in total, equating to 3.9 ± 1.3 years for the rapid decliners, and 3.4 ± 1.4 years for those with non-rapid decline, and there was no significant difference in the follow-up duration between the two groups.

#### 2.2.2. Frequency of Exacerbations

According to the GOLD report [23], patients with COPD were divided into groups with infrequent and frequent exacerbator (≥2 acute exacerbations/year or ≥1 severe acute exacerbation/year). In this study, we obtained the patients’ medical history a year before we obtained the PBMC data; there were 12 frequent exacerbators and 40 non-frequent exacerbators.

#### 2.2.3. Eosinophilic COPD

Eosinophilic COPD is defined as a blood eosinophil count >300 cells/μL based on treatment considerations from the GOLD report recommendations [23]. There were 7 eosinophilic and 47 non-eosinophilic COPD patients in this study.

#### 2.2.4. COPD with Emphysema

Emphysema is represented by the percentage of low attenuation areas (LAAs) calculated using quantitative computed tomography (CT) via a density mask method, with a threshold of ≤ 950 Hounsfield units (HU). We defined the mild (LAA ≤ 15%) and severe (>15%) emphysema groups based on a previous study [24]. There were 17 and 15 patients with mild and severe emphysematous COPD in our study, respectively.

### 2.3. Peripheral Blood Mononuclear Cell (PBMC) Isolation

After obtaining informed consent, 30 mL of whole blood from 29 normal subjects and from 54 patients with COPD with different phenotypes was collected. The plasma was then collected from the whole blood with an anticoagulant by centrifuging 2000× *g* for 10 min at 4 °C before PBMC separation and stored at −80 °C until use. The rest of the blood cells were used for PBMC isolation and the following in vitro experiments. The PBMCs were isolated from the whole peripheral blood using gradient centrifugation (Lymphoprep™, Catalog #07801, StemCell Technologies Inc., Vancouver, BC, Canada).

### 2.4. Cell Culture and Stimulation

The PBMCs were resuspended in 2 mL RPMI in 6-well plates for 24 h prior to experimentation. To determine the effect of PD-L1+ cDC1 on lymphocyte proliferative status, the PBMCs were stimulated with an anti-CD3 (Catalog #300314; Biolegend, San Diego, CA, USA) (10 µg/mL) or anti-CD3+ elastin peptide (Catalog #QP45; Elastin Products, Owensville, MO, USA) (30 ng/mL) with or without anti-PD-L1 (Catalog #329710, Biolegend, San Diego, CA, USA) (2 µg/mL) for 96 h, and then an anti-Ki-67 (Catalog #350506, Biolegend, San Diego, CA, USA) antibody was used as a proliferation marker under the indicated culture conditions. The levels of PD-L1, Ki-67 and the markers of the DC and T cell subsets were examined using flow cytometry. To examine the cytokine levels, culture supernatants were harvested from the PBMCs of the healthy controls and patients with COPD stimulated with the anti-CD3 antibody (10 μg/mL) or anti-CD3 combined with the elastin peptide (30 ng/mL), with or without anti-PD-L1 (2 μg/mL) for 48 h, and then measured using ELISA assays.

### 2.5. Flow Cytometry

The PBMCs were stained according to the manufacturer’s recommendations. To measure the expression of PD-L1, 5 × 10^5^ cells were incubated for 30 min at 4 °C with specific mouse anti-human monoclonal antibodies conjugated with fluorochromes. Anti-human HLA-DR-APC/Fire™ 750 (Catalog #307658, Biolegend, San Diego, CA, USA) and anti-lineage-FITC (Catalog #348701, Biolegend, San Diego, CA, USA) were used to define the DCs. Anti-human 11c-Alexa Fluor^®^ 700 (Catalog #56-0116-42; ThermoFisher Scientific, Waltham, MA, USA) in combination with anti-human CD141-PE-Cy7 (Catalog #344110, Biolegend, San Diego, CA, USA), anti-human CD1c-Alexa Fluor^®^ 647 (Catalog #565048; BD, Franklin Lakes, NJ, USA), and anti-human CD123-PerCP/Cyanine5.5 (Catalog #45-1239-42, ThermoFisher Scientific, Waltham, MA, USA) were used to define cDC1, cDC2, and pDC, respectively. Anti-CD4-BV786 (Catalog #344642, Biolegend, San Diego, CA, USA) and anti-CD8- BV711 (Catalog #301044, Biolegend, San Diego, CA, USA) were used to identify the CD4 and CD8 T cell subsets. The data were analyzed with FlowJo™ Software v10.0. More details of the gating strategies for the blood dendritic cells and T lymphocytes are provided in the Appendix A. The method to determine PD-L1 expression is based on the isotype of PD-L1, and this was determined using histogram analysis, as shown in the Appendix A.

### 2.6. ELISA

The levels of IFN-γ (Catalog #DY285B-05), IL-4 (Catalog #DY204-05), IL-17A (Catalog #DY317-05), and IL-10 (Catalog #DY217B-05) in the culture supernatants were measured with commercial ELISA kits (R&D Systems; Minneapolis, MN, USA). Briefly, 100 μL of the sample or standards were added per well containing a diluent and were incubated at 24 °C for 2 h. This was followed by carefully washing them with cold 1× PBS 3 times, the addition of 100 µL substrate solution to each well, and incubation at 24 °C for 20 min in a dark room. Thereafter, 50 μL of stop solution was added per well, and the optical density determined immediately using a microplate reader set to 450 nm.

### 2.7. Statistical Analysis

All statistical analyses were performed using GraphPad Prism V5.0. The results are expressed as means ± standard deviation of the assays performed in at least three independent experiments. Comparison between 2 groups of normally distributed data was conducted using a two-tailed unpaired *t*-test; otherwise, Mann–Whitney U test was used. For groups of 3 or more, a one-way ANOVA test followed by Tukey’s post hoc tests were used. The non-normally distributed data were analyzed using the Kruskal–Wallis test and Dunn’s multiple comparison test. Spearman’s rank correlation aided the analysis of correlations between the variables. A *p*-value < 0.05 was considered to be statistically significant.

## 3. Results

### 3.1. PD-L1 Expression and Distribution of DCs and Subsets in Normal Participants and Those with COPD

During the initial screening of PD-L1 expression by variable PBMCs, the number of PD-L1-expressing dendritic cells in the patients with COPD decreased compared with those in the normal subjects. After expanding by n numbers, we confirmed that the ratio of PD-L1^+^ cDC1 was lower in the patients with COPD (*n* = 54) than in the normal subjects (*n* = 29) (13.7 ± 13.7% vs. 32.7 ± 33.1%, respectively; *p* = 0.03, Figure 1A). In contrast, the PD-L1 expression by both cDC2 and pDC was similar between the patients with and without COPD. The reduction in PD-L1^+^ cDC1 was not attributable to a change in the total cDC1and DCs (Figure 1B). The ratio of PD-L1 expressed by cDC1 is significantly reduced in the patients with COPD. In addition, the digital microscopy images show that PD-L1 fluorescent staining (shown as green color) in cDC1 is more reduced in the patients with COPD than in the normal subjects (Figure 1C); therefore, we confirmed decreased PD-L1 expression in cDC1 in the patients with COPD.

### 3.2. Decrease in PD-L1^+^ cDC1 Total Was Associated with Rapid Lung Function Decline in COPD

Next, we asked whether the decrease in PD-L1^+^ cDC1 was linked to any clinical characteristics, particularly the distinct phenotypes of COPD. We found that the proportion of PD-L1^+^ cDC1 was significantly lower in the patients with COPD with rapid lung function decline compared with those without a rapid decline in lung function (4.6 ± 4.7% vs. 15.5 ± 18.6%, respectively; *p* = 0.02, Figure 2A). This association was specific to the subtype cDC1, which was selectively decreased in PD-L1 in this subtype of DC (Figure 1A). The PD-L1 expressions by DCs were compared among the patients with COPD at different GOLD stages and the normal subjects. The results showed that a statistically significant reduction in the number of PD-L1^+^ cDC1 cells was only seen in the patients with GOLD stage II COPD and not in the normal subjects (10.6 ± 10.8% vs. 32.5 ± 33.1%, respectively, *p* = 0.01, Figure 2B). These observations were in line with previous studies on the trajectory of lung function, showing the fastest decline in lung function at the same stage of the disease [25]. There was no significant difference in the expression levels of total PD-L1 among the three subtypes of DCs in the phenotypes examined, including frequent exacerbators, eosinophilic, and emphysema (Figure 2C–E). Thus, PD-L1 expression by cDC1 is distinctively associated with the rapid decline in lung function.

### 3.3. PD-L1^+^ cDC1 Was Negatively Correlated with CD4^+^ T Cells

Based on our previous results, we further investigated the mechanism through which the reduction in the ratio of PD-L1^+^ cDC1 leads to a loss of lung function. As both the CD4^+^ and CD8^+^ T cells are implicated in the pathogenesis of COPD, we asked whether PD-L1 expression by cDC1 was associated with the ratio of both T cells in the PBMCs. We observed a significantly negative correlation between the ratio of PD-L1^+^cDC1 with that of the CD4^+^ T cells in the PBMCs from the patients with COPD (*r* = −0.37, *p* = 0.049, Figure 3A). Consistent with our previous findings, there was no association of cDC2 and pDC with the CD4^+^ T cells. By contrast, and unexpectedly, we did not see links between either subtype of DC and the CD8^+^ T cells (Figure 3B). To support this finding, we used the NCBI GEO database accession number GSE56766 and analyzed the relationship between *PD-L1* and *CD4* or *CD8* gene expression. This dataset included microarray data from the whole blood transcriptomics of 49 patients with COPD. In agreement with our FACS findings, the gene expressions of *PD-L1* and *CD4* but not *CD8* were negatively correlated (*r* = −0.43, *p* = 0.002, Figure 3C).

### 3.4. PD-L1 Was Involved in Suppression of CD4^+^ T Cell Proliferation and Th17 Cell Differentiation

To understand the functional role of PD-L1 in the activation and differentiation of CD4^+^ T cells, we conducted a series of in vitro studies using PBMCs from the normal subjects and patients with COPD. The previous reports indicated the involvement of an elastin-specific T cell response in COPD [14]. Thus, we used the elastin peptide as a stimulant. In the absence of elastin, the specific anti-PD-L1 blocking antibody had no effect on the proliferation of the CD3-activated CD4^+^ T cells, measured by the proliferation marker Ki-67. In the presence of elastin, CD3 stimulation induced the proliferation of CD4 T^+^ cells at a level similar to CD3 stimulation only. However, the PD-L1 blockade robustly enhanced proliferation (14.9 ± 15.4% vs. 31.2 ± 22.3%, respectively; *p* = 0.04, Figure 4). In the PBMCs from the patients with COPD, CD3 plus elastin co-stimulation induced the stronger proliferation of CD4^+^ T cells compared with those from the normal subjects (35.2 ± 21.5 vs. 14.9 ± 15.4%, respectively, *p* = 0.03, Figure 4). Interestingly, in the absence of the PD-L1 blocking antibody, proliferation in COPD occurred similarly to that in the normal subjects with PD-L1 blockade. These data suggest that PD-L1 is functionally involved in the tolerance of the CD4 T cell response to elastin. Reduced PD-L1 expression by cDC1 might impair this tolerance, leading to autoimmunity in COPD. Because CD4^+^ T cell dysregulation in COPD involves the polarization of distinct subtypes [26], we further tested their association with the expression of PD-L1. Again, we used the NCBI GEO database accession number GSE56766 to analyze the relationship between the gene expression level of *PD-L1* and the subtypes of CD4 T cells, including Th1, Th2, Th17, and Treg cells. The results showed that only the expression of *RORC*, a master transcription factor driving the Th17 cells, was significantly negatively correlated to PD-L1 (*r* = −0.33, * *p* = 0.02, Figure 5A).

In in vitro experiments, it has been shown that PD-L1 blockade augmented the production of IL-17A in both the CD3- (156.3 ± 54.7 vs. 108.2 ± 45.0 pg/mL, *p* = 0.03, Figure 5B) and CD3/elastin-stimulated (148 ± 64.9 vs. 106.5 ± 43.2 pg/mL, respectively; *p* = 0.03, Figure 5B) PBMCs from normal subjects. By contrast, there was no significant change in IFN-γ, IL-4, and IL-10 after the PD-L1 blockade. To further confirm the contribution of cDC1 in IL-17A production and the tolerogenic roles of PD-L1^+^ cDC1, the cDC1 cells were removed from the PBMCs via cell sorting. During the deletion of cDC1, neither CD3 nor CD3/elastin was able to stimulate IL-17A production (Figure 5C). Although CD3 stimulation seemed to induce weak IL-17A production, it was not statistically significant. Importantly, the PD-L1 blockade did not affect the production of 17A. Taken together, PD-L1 in the PBMCs is functionally related to suppression of CD4^+^ T cell proliferation and Th17 differentiation.

## 4. Discussion

Our study demonstrated that PD-L1^+^ cDC1 was reduced in the patients with COPD and was associated with a rapid decline in lung function. This might be caused by the loss of PD-L1-mediated tolerance, leading to the activation of CD4^+^ lymphocytes with Th17 polarization. Our study suggests a tolerogenic role of PD-L1^+^ cDC1 and its clinical relevance in COPD.

PD-L1 is constitutionally expressed on antigen-presenting cells. PD-L proteins on the tolerogenic DCs limit self-reactive T cell activation to maintain peripheral tolerance and prevent autoimmunity [27]. Our data revealed that PD-L1 expression was specifically reduced in the circulating cDC1 of the patients with COPD. This reduction was negatively correlated with the CD4^+^ T cells, suggesting a tolerogenic role of PD-L1^+^ cDC1. Importantly, we found a link between PD-L1^+^ cDC1 reduction and those with a rapid decline in lung function. The unique role of the PD-L1^+^ cDC1 among the three subtypes of DCs in COPD was consistent across variable observations, including its ratio and relationship with clinical phenotypes and the CD4^+^ T cells, which makes these findings more convincing. Our results suggest that loss of PD-L1^+^ cDC1-mediated tolerance may perpetuate inflammation in response to self-antigens, such as elastin.

One previous report demonstrated that the amount of PD-L1 decreased in the circulating pDCs from patients with COPD [28], which was not seen in our study. This could be due to the different cohorts of patients studied, particularly with different ethnic groups. It is not clear whether PD-L1 in cDC1 was reduced in this report. In addition to the DCs, the decreased virus-induced expression of PD-L1 on the COPD macrophages was reported with a corresponding increase in IFN-g levels [29].

In contrast to the clear association with the rapid decline in those with COPD, the PD-L1^+^ cDC1 level was only statistically reduced in the patients with GOLD stage II disease. Nevertheless, this finding is consistent with the fastest decline in lung function at this stage of COPD [23,25]. It is likely that loss of PD-L1^+^ cDC1-mediated tolerance becomes severe at this stage and induces an aberrant adaptive immune response to self-antigens [30], leading to airway remodeling and lung damage. In the later stages, PD-L1 should be induced by persistent inflammation; however, this did not occur to the same extent as in our observation. It is not clear why the level of PD-L1 is reduced in cDC1. One possibility is that it materialized through an epigenetic mechanism. For example, oxidative stress in COPD could increase the quantity of microRNA-34a (miR-34a) and downregulate the expression of PD-L1 [31,32]. Alternatively, we could not exclude the possibility that PD-L1^+^ cDC1 constitutes a specific lineage of DCs. One incidental but interesting finding in our study is the significantly decreased percentage of cDC2 in the PBMCs in the patients with COPD. Unlike cDC1, cDC2 is more heterogeneous [33] but the limited number of recruited COPD patients in this study has confounded further interpretations. However, Tsoumakidou, M. et al. showed that pulmonary CD1c^+^ (cDC2) could exert a tolerogenic function among the regulatory T cells in patients with COPD [34]. Therefore, the decreased percentage of cDC2 in the PBMCs in the patients with COPD in our study may drive persistent inflammation. This is consistent with our hypothesis.

Two lines of evidence support a functional role of PD-L1 in the suppression of Th17 in the PBMCs: the negative correlation between RORC and PD-L1 gene expression in COPD, and the anti-PD-L1 enhancement of IL-17A production in the normal PBMCs. Thus, PD-L1 in PBMCs plays the role of Th17 tolerance in the steady state. In COPD, reduced PD-L1^+^ cDC1 expression might lead to the loss of this tolerance and Th17 inflammation. In a mouse model of cigarette smoke exposure, MMP-12-generated elastin fragments could serve as an autoantigen to induce autoimmune processes by IL-17A [14]. In line with this, decreased PD-L1^+^ cDC1 expression in COPD might lead to the loss of tolerance and cause a Th17-skewed CD4^+^ T cells reaction in response to such self-antigens. Indeed, when cDC1 was removed from the PBMCs, the PD-L1 blockade failed to enhance Th17 immunity, supporting this scenario; however, conventionally, cDC2 induces Th17 polarization [35]. Nevertheless, the functions of cDC1 and cDC2 seem to be plastic and can drive T cell-specific immunity according to different environmental stimuli. It was shown in a mouse model that intranasal immunization with adenosine diphosphate (ADP)-ribosylating adjuvant modified cDC1 cells to effectively prime the Th17 cells [36]. More direct evidence has shown that vitamin D receptor (VDR)-KO mice increased the expression of Th17 cells, and this was associated with a reduction in tolerogenic CD103^+^ dendritic cells [37]. In addition, a mouse CD103^+^ subset of tolerogenic gut DCs acted directly on the T cells to reduce their capacity for IL-17 production by producing thymic stromal lymphopoietin (TSLP) [38]. As CD103^+^ DCs are the mouse equivalent of human cDC1, it is tempting to suggest that PD-L1^+^ cDC1 is tolerogenic to Th17.

Therefore, this research provides a new immunopathogenic mechanism for COPD, in which reduced PD-L1^+^ cDC1 expression may cause the loss of immune tolerance to self-antigens, induce Th17 immunity, and lead to a decline in lung function. However, there are some limitations to our study. Firstly, those in the control group were not specifically matched by age and smoking status with the patients with COPD. Age and smoking will affect the expression of PD-L1 in patients with conditions such as obstructive sleep apnea [39] and rheumatoid arthritis [40] but it is not clear whether this affects normal subjects. Secondly, though the prevalence of males with COPD is still higher than that of females [41], the significant difference in the male/female ratio in the COPD and control groups is another limitation, despite the clinical symptoms and biological mechanisms having been reviewed in recent years [42]. Thirdly, normal participants were recruited on the basis of observing no past underlying diseases in their medical history. We did not measure pulmonary functional parameters in this group. Fourthly, regarding the study of mechanism, we did not purify the cDC1 and T cells for co-culture stimulation, mainly because the content of cDC1 in the blood is too small, so the results of the PD-L1 blocking experiments on the PBMCs may have been caused by other immune cells. Finally, the conclusion that decreased PD-L1^+^ cDC1 expression is linked to the COPD phenotype of rapid lung function decline and the CD4 T cells may not be solid enough due to the small sample size. Nevertheless, this novel concept is worth exploring further.

## 5. Conclusions

In conclusion, we found reduced circulating PD-L1^+^ cDC1 in COPD, which was linked to rapid lung function decline. We proposed a theory in which PD-L1^+^ cDC1 is a tolerogenic DC controlling the Th17 response to self-antigens under steady conditions. In COPD, the reduction in PD-L1^+^ cDC1 expression causes a loss in self-tolerance and enhances the susceptibility to self-antigens, particularly from apoptotic cells, leading to Th17-skewed inflammation. PD-L1^+^ cDC1 might provide a novel treatment strategy for COPD.

## Figures and Tables

**Figure 1 cells-13-00878-f001:**
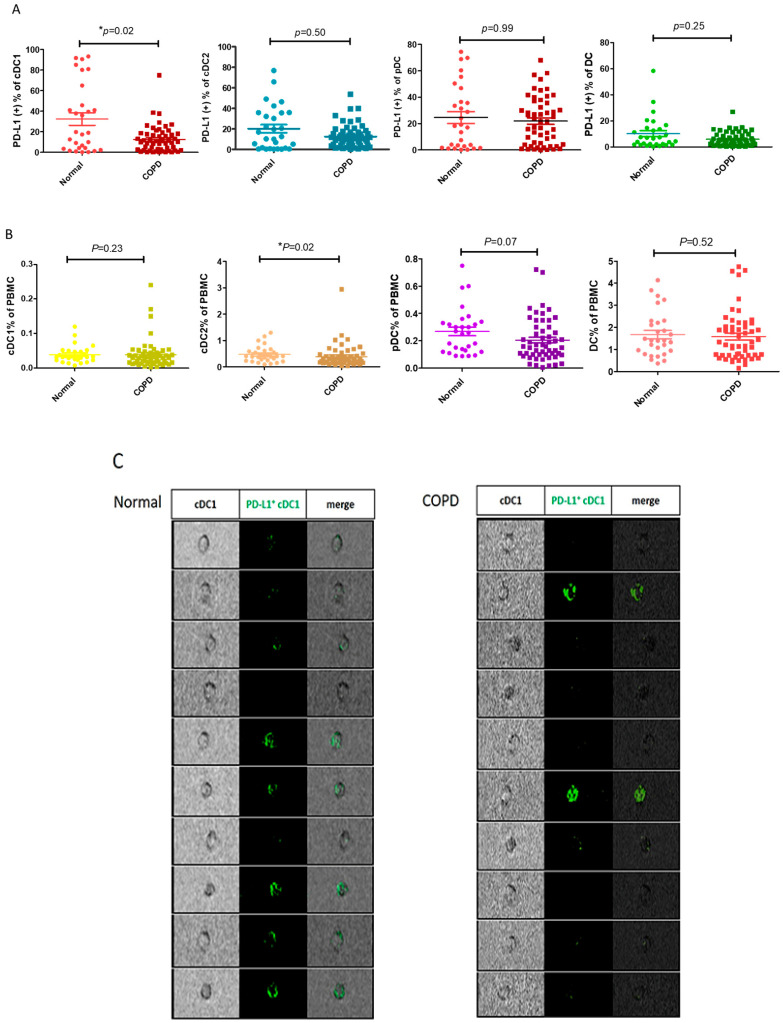
PD-L1 expression and distribution among the DCs and subsets in the normal participants and those with COPD. Flow cytometric analysis showing PD-L1 expression by the DCs and subsets in the normal patients (*n* = 29) and those with COPD (*n* = 54). The DCs are defined as HLA-DR^+^lin^−^, whereas the plasmacytoid dendritic cells (pDCs) are defined as CD123^+^HLA^−^DR^+^lin^−^. Conventional type 1 DC (cDC1) and conventional type 2 DC (cDC2) are defined as CD141^+^HLA-DR^+^lin^−^ and CD1C^+^HLA-DR^+^lin^−^, respectively. The PBMCs were stained directly with the indicated antibodies for analysis. (**A**) The percentage of PD-L1^+^ DCs and subsets in the normal subjects and those with COPD are shown. (**B**) The ratios of DCs and subsets to PBMCs are indicated in the normal participants and those with COPD. (**C**) Representative images of PD-L1^+^ cDC1 (green) from the normal participants and those with COPD, showing spatial resolution and quantitative morphology, using digital microscopy. The data are shown as means ± SEM. A Mann–Whitney U test was performed for analyzing data of the normal and COPD groups. * *p* < 0.05.

**Figure 2 cells-13-00878-f002:**
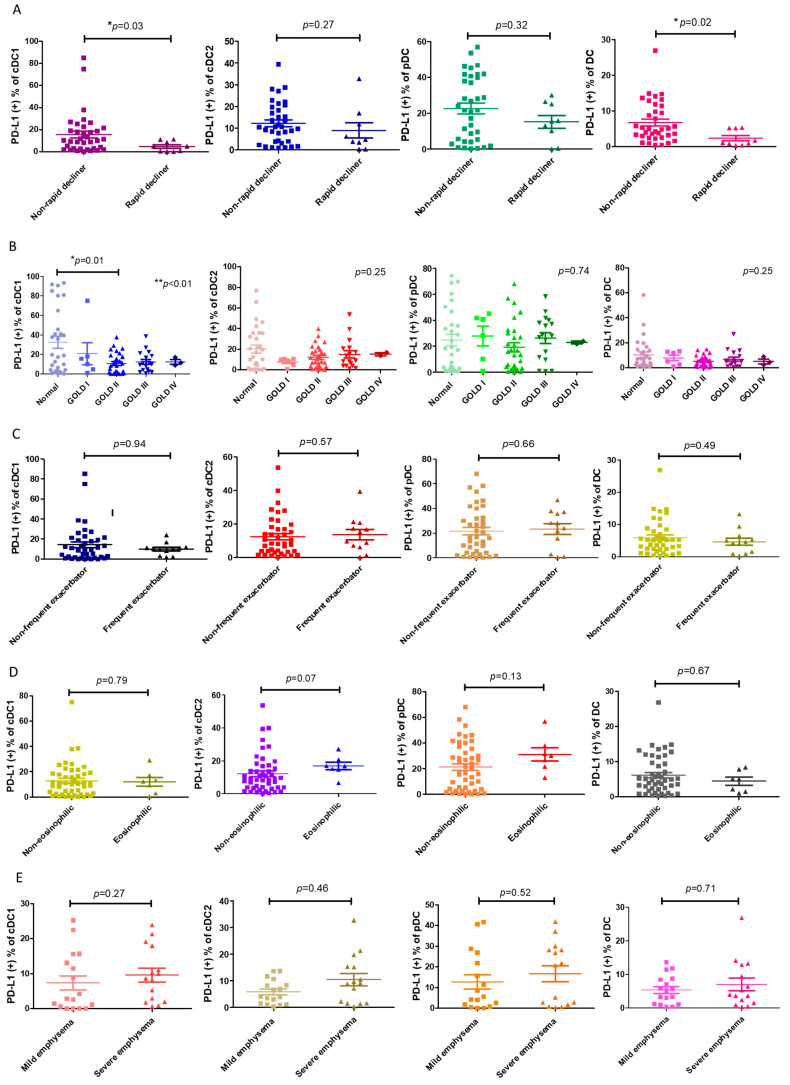
The association of PD-L1^+^ DCs and their subsets with COPD phenotypes. (**A**) A loss of forced expiratory volume in one second (FEV1) ≥ 60 mL/year was defined as a rapid decline in lung function in COPD. The ratios of PD-L1^+^ expression by DCs and subpopulations of rapidly declining patients with COPD (*n* = 9) and non-rapidly declining patients (*n* = 35) are shown. (**B**) The percentages of PD-L1^+^ expression by the DCs and subsets of normal (*n* = 29) and GOLD stage I–IV participants with COPD (*n* = 54) are shown, including 6 stage I, 28 stage II, 17 stage III, and 3 stage IV patients. (**C**) The percentage of patients with COPD with frequent or infrequent exacerbations, showing PD-L1^+^ expression by the DCs and subsets. According to the number of acute exacerbations per year, the patients with COPD were divided into frequent (*n* = 12) (≥2 acute exacerbations/year or ≥1 severe acute exacerbation/year) and infrequent acute exacerbation groups (*n* = 40). (**D**) The percentage of PD-L1^+^ expression by the DCs and subsets of 7 eosinophilic or 47 non-eosinophilic COPD patients. Eosinophilic COPD is defined as a blood eosinophil count >300 cells/μL. (**E**) Emphysema is represented by the percentage of low attenuation areas (LAAs) calculated using quantitative computed tomography (CT) via a density mask method with a threshold of ≤−950 Hounsfield units (HU). The PD-L1^+^-expressing cells are shown on the DCs and subsets of patients with mild (LAA ≤ 15%, *n* = 17) and severe (>15%, *n* = 15) emphysema. The data are shown as means ± SEM. For each COPD phenotype, a Mann–Whitney U test was used to compare the data of the two groups. If there were more than 3 groups, we first used ANOVA analysis, followed by Tukey’s post hoc tests. * *p* < 0.05; ** *p* < 0.01.

**Figure 3 cells-13-00878-f003:**
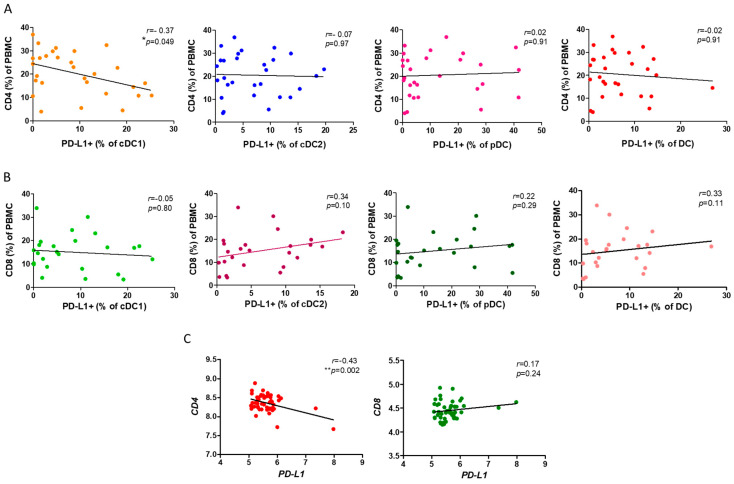
The association of PD-L1^+^ DCs and subsets with CD4^+^ or CD8^+^ T lymphocytes of PBMCs. (**A**) The flow cytometry analysis of the correlation between the PD-L1^+^ DCs and the subsets and CD4^+^ T in the PBMCs of 29 COPD patients. (**B**) Flow cytometric analysis demonstrated a correlation between the PD-L1^+^ DCs and subsets and CD8^+^ T cells in the COPD PBMCs. (**C**) The expression of CD4 and CD8 genes in the whole blood of 49 COPD patients (GSE56766 dataset) was correlated with PD-L1 expression. Spearman’s rank correlation was used for the variables. * *p* < 0.05; ** *p* < 0.01.

**Figure 4 cells-13-00878-f004:**
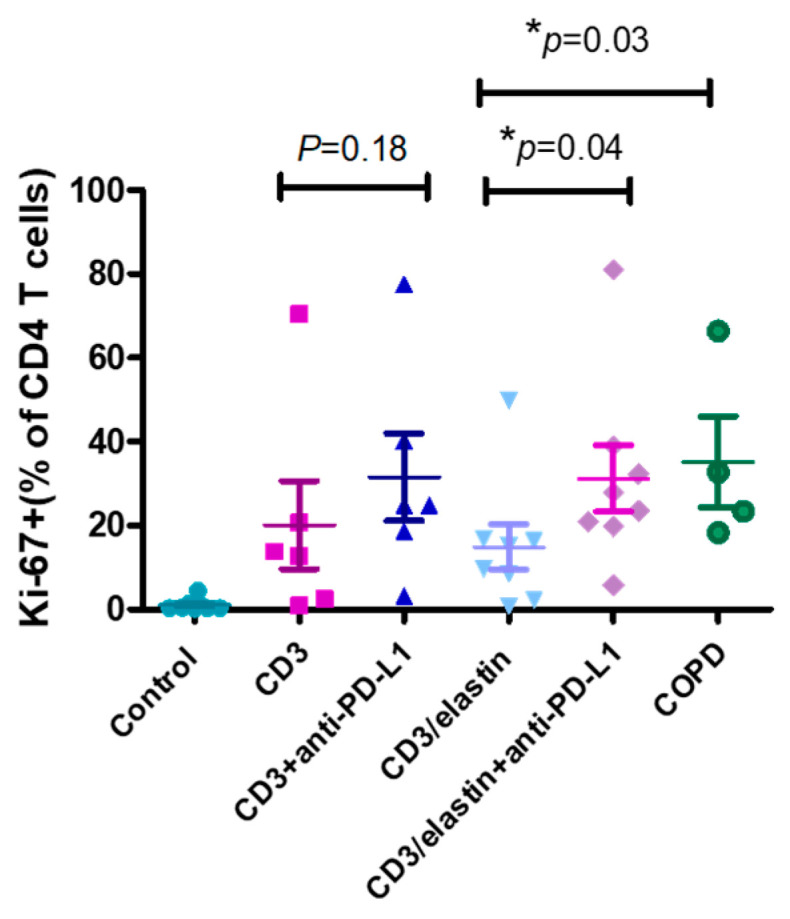
PD-L1 is involved in inhibiting CD4 T cell proliferation. Representative proliferation marker (Ki67) for CD4^+^ cells from normal subjects (*n* = 6–8) and patients with COPD (*n* = 4) with and without anti-PD-L1 blocking antibodies after in vitro stimulation with anti-CD3 and anti-CD3/elastin peptides (30 ng/mL). Mann–Whitney U test was used to compare the data of the two groups under the stimulated condition. * *p* < 0.05.

**Figure 5 cells-13-00878-f005:**
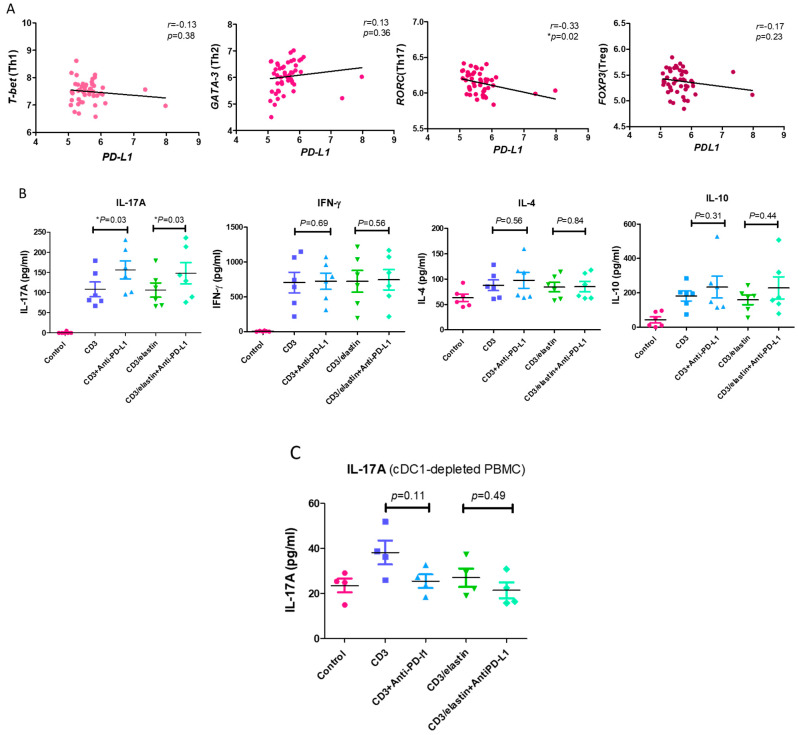
PD-L1 is involved in inhibiting the differentiation of Th17 cells. (**A**) The expression levels of *T-bet* (represented as Th1), *GATA-3* (represented as Th2), *RORC* (represented as Th17), and *FOXP3* (represented as Treg) genes in the whole blood of 49 COPD patients (GSE56766 dataset) were correlated with the expression of PD-L1. (**B**) The Th1 (interferon-gamma, IFN-gamma), Th2 (interleukin 4, IL-4), Th17 (interleukin 17A, IL-17A), and Treg cytokine (interleukin 10, IL-10) levels in the CD4^+^ subset cells from 6 normal participants stimulated with a CD3 or CD3/elastin peptide alone or with an anti-PD-L1 blocking antibody were compared with those of the vehicle controls. These cytokine levels produced by the CD4^+^ subset cells in the normal participants were measured using ELISA. (**C**) The comparison of Th17 cytokines (interleukin 17A, IL-17A) between the normal subjects stimulated with a CD3 or CD3/elastin peptide alone or with an anti-PD-L1 blocking antibody (*n* = 4) from the cDC1-depleted PBMCs. Mann–Whitney U test was used to compare the data of the two groups under the stimulated condition. * *p* < 0.05.

## Data Availability

The data used in the current study are all contained in the manuscript and may be obtained upon reasonable request from the corresponding author.

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
