# Peer review of "Reduced Tolerogenic Program Death-Ligand 1-Expressing Conventional Type 1 Dendritic Cells Are Associated with Rapid Decline in Chronic Obstructive Pulmonary Disease"

_cells, 2024, doi:10.3390/cells13100878_

Round 1

Reviewer 1 Report

Comments and Suggestions for Authors

The authors present a manuscript regarding assessment of PDL1 expression on subtypes of peripheral blood dendritic cells (DC) among COPD patients and "healthy" controls. They report lower proportions of PDL1+ DC among COPD patients, and an especially lower proportion of such cells in COPD patients with a profound annual FEV1 decline (rapid decliners).

However, the study methodology is not presented in a coherent manner, no hypothesis is presented to describe grouping or categorization of participants, and some of the laboratory methods are not considered appropriate for evaluating DC from peripheral blood. 

In addition, major language revision is necessary. 

In the current state I am sorry but the manuscript is not ready for publication. 

Comments on the Quality of English Language

Not comprehensible in some sections, wrong grammar

Reviewer 2 Report

Comments and Suggestions for Authors

The manuscript by Chen et al. have studied PDL1 expressing cDCs during COPD. Though the study has skewed ratios of number of patients, sex percentiles and ages between the healthy and diseased groups, it details the objectives for this study. Comments are listed below:

Comments:

Is there a reason why the levels of IFN-γ, IL-4, IL-17A, and IL-10 were considered to be measured amongst other cytokines? Please specify?

Was B cell, NK cells and monocyte populations looked into during flow analysis? Were there any differences in their PDL1 expression between normal and COPD subjects?

Figure 2 a. Non-rapid decliner mentioned as non-rpide decliner in figures. Please correct.

Add a table for Figure 2 showing number, percentages and p values between COPD phenotypes for better visualization of the numbers.

Please elaborate on “PD-L1+ cDC1 might provide a novel treatment strategy for COPD.” Are there therapeutic methods being studies on these subsets for treatment strategies for other diseases? How do authors think their study will contribute to the current therapeutic strategies?

Comments on the Quality of English Language

Their are few mistakes in the use of English and must be corrected thoroughly.

Reviewer 3 Report

Comments and Suggestions for Authors

The current manuscript from Dr. Lee’s research group addresses the tolerogenic role of PD-L1+dendritic cells (DCs) in normal and COPD patients.

Although the observations are interesting, but relative impact of the study is very limiting. Furthermore, the study design only explored the topic superficially, and more comprehensive research is required to understand the underlying mechanisms. Also, authors may adopt a different approach to address the research question, as indicated in the review. There are too many gaps in the study design, experimentation, and data analysis strategies.

Suggestions

1.      A detailed methods section on the culture of PBMC must be provided in the manuscript. This is very important as the in vitro experiments were carried out only on a few subjects, and probably on different days.

2.      The supplementary table 1 shows the characteristics of normal and COPD participants. There are a few concerns here. First, no pulmonary function parameters have been provided for the control subjects. Second, the significant difference between the two groups in terms of male/female ratio is a critical issue with the study. Authors are suggested to separate the male subjects from the female subjects and report the data for each sex separately. The number of very few female subjects will make it hard to conclude any meaningful conclusion, but the results from the male subjects will be informative.

3.      The details of statistical analysis need to be expanded. Especially, whether any post hoc test was used or not. Please include the statistical analysis used for each figure in the legends.

4.      Figure 1B, the percentage of cDC2 in PBMCs was significantly decreased in COPD patients, compared to normal subjects. Please expand discussion on this.

5.      The in vitro experiment described in the manuscript only includes some COPD patients. Authors must provide details of the characteristics of these patients.

6.      The observations from figure 1C on representative images of PD-L1+ cDC1 from normal and COPD participants are only useful if a quantitative assessment is provided.

7.      The limitations section must be expanded.

8.      Supplementary figures S1 and S2 have not been discussed in the manuscript.

9.      The manuscript requires thorough editing for language use.

Other comments

1.      The first few lines of the results section “PD-L1+ cDC1 was decreased in PBMCs of COPD patients” is unnecessary. Authors may report the results of the study.

2.      Figure 2A, please check the spelling of “rapid”.

3.      Please provide the catalogue numbers of all reagents.

4.      Part of the observations from the study were derived from analyzing NCBI GEO database accession number GSE56766; this is not reflected in the abstract.

Comments on the Quality of English Language

Must be improved; currently far away from a publishable standard.

Reviewer 4 Report

Comments and Suggestions for Authors

Kuan-Yuan Chen et al. determined the role of PD-L1+dendritic cells (DCs) and their subtypes in relation to specific T cell immunity and clinical rapid lung function decline of COPD through flow cytometry, cell culture, and ELISA.

The manuscript has merits, and I do not doubt that experts in the field will very well accept it. I have no major comments except that they should read the manuscript thoroughly for some editing mistakes.

The introduction gives the necessary information to understand the manuscript.

The methodology is sufficient to be reproducible for the interested reader.

The figures and tables are understandable, and I am sure that the quality of the figures and tables will increase in definition when the original files are uploaded.

The conclusion is adequate for the findings.

Round 2

Reviewer 2 Report

Comments and Suggestions for Authors

The comments have been well-addressed by the authors.

Reviewer 3 Report

Comments and Suggestions for Authors

Authors tried their best to address the suggestions.

Comments on the Quality of English Language

Needs editing.

Reviewer 4 Report

Comments and Suggestions for Authors

Kuan-Yuan Chen et al. determined the role of PD-L1+dendritic cells (DCs) and their subtypes about specific T cell immunity and clinical rapid lung function decline of COPD through flow cytometry, cell culture, and ELISA.

I have no major or minor comments; the authors corrected my previous comments point by point.